# Label Decoupling and Reconstruction: A Two-Stage Training Framework for Long-tailed Multi-label Medical Image Recognition

Jie Huang
Wenzhou University
Wenzhou, China
23451350016@stu.wzu.edu.cn

Zhao-Min Chen*
Xiaoqin Zhang*
Wenzhou University
Wenzhou, China
chenzhaomin123@gmail.com
zhangxiaoqinnan@gmail.com

Yisu Ge
Wenzhou University
Wenzhou, China
ysg@wzu.edu.cn

Lusi Ye
The First Affiliated Hospital of
Wenzhou Medical University
Wenzhou, China
yelusi@wmu.edu.cn

Guodao Zhang
Hangzhou Dianzi University
Hangzhou, China
guodaozhang@hdu.edu.cn

Huiling Chen
Wenzhou University
Wenzhou, China
chenhuiling@gmail.com

## Abstract

Deep learning has made significant advancements and breakthroughs in medical image recognition. However, the clinical reality is complex and multifaceted, with patients often suffering from multiple intertwined diseases, not all of which are equally common, leading to medical datasets that are frequently characterized by multi-labels and a long-tailed distribution. In this paper, we propose a method involving label decoupling and reconstruction (LDRNet) to address these two specific challenges. The label decoupling utilizes the fusion of semantic information from both categories and images to capture the class-aware features across different labels. This process not only integrates semantic information from labels and images to improve the model's ability to recognize diseases, but also captures comprehensive features across various labels to facilitate a deeper understanding of disease characteristics within the dataset. Following this, our label reconstruction method uses the class-aware features to reconstruct the label distribution. This step generates a diverse array of virtual features for tail categories, promoting unbiased learning for the classifier and significantly enhancing the model's generalization ability and robustness. Extensive experiments conducted on three multi-label long-tailed medical image datasets, including the Axial Spondyloarthritis Dataset, NIH Chest X-ray 14 Dataset, and ODIR-5K Dataset, have demonstrated that our approach achieves state-of-the-art performance, showcasing its effectiveness in handling the complexities associated with multi-label and long-tailed distributions in medical image recognition.

___

*Corresponding author.

___

## CCS Concepts

- **Computing methodologies → Biometrics**;

## Keywords

Long-tail, Multi-label, Medical image recognition, Label decoupling, Gaussian distribution reconstruction

**ACM Reference Format:**
Jie Huang, Zhao-Min Chen, Xiaoqin Zhang, Yisu Ge, Lusi Ye, Guodao Zhang, and Huiling Chen. 2024. Label Decoupling and Reconstruction: A Two-Stage Training Framework for Long-tailed Multi-label Medical Image Recognition. In *Proceedings of the 32nd ACM International Conference on Multimedia (MM '24), October 28-November 1, 2024, Melbourne, VIC, Australia.* ACM, New York, NY, USA, 9 pages. https://doi.org/10.1145/3664647.3680606

## 1 Introduction

In recent years, deep learning has achieved significant accomplishments in the field of computer-assisted diagnosis [6, 9, 41, 45]. Medical image recognition, in particular, plays a crucial role in the clinical diagnosis of diseases. The ability to automatically identify specific diseases or conditions in medical images can significantly assist physicians in clinical diagnosis, reduce their workload, and improve diagnostic efficiency. However, in real-world clinical applications, medical images (such as CT scans, X-rays, etc.) often contain multiple diseases, thereby inevitably exhibiting multi-label characteristics. Additionally, due to the varying incidences of diseases, medical image datasets typically show a long-tail distribution, where common cases (head categories) dominate a large number of samples while rare cases (tail categories) are represented in smaller quantities. Therefore, the medical image recognition tasks in real scenarios are confronted with two significant challenges: multi-label and long-tail distribution (refer to Fig. 1). Without proper handling, trained models may develop a bias towards head categories, presenting substantial challenges to the task.

Considerable efforts have been made towards multi-label medical image recognition. Among the existing works on medical image classification, Zhang et al.[49] introduced a triplet attention

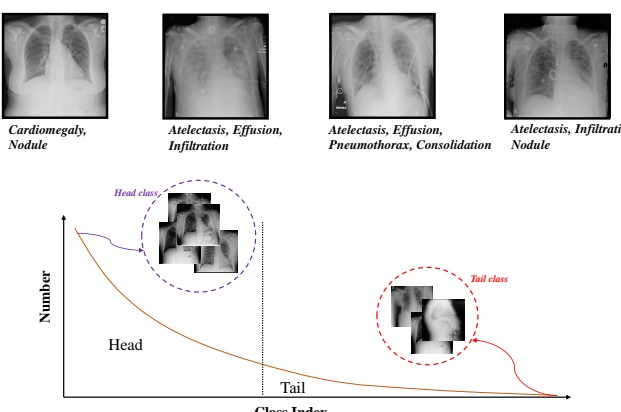

**Figure 1: The top part depicts the multi-label scenario of medical images, where each medical image corresponds to multiple labels of pathological features. The bottom part shows the long-tail distribution of medical images, indicating an extremely imbalanced distribution within the medical image dataset.**

mechanism combined with contrastive learning to mine effective information from medical images for learning high-quality label embeddings for all disease labels. He et al.[16] employed a knowledge distillation-based approach to enhance the diagnostic performance of multi-label medical image diseases. Zhou et al.[53] addressed the challenge of extending single-label to multi-label scenarios through a cross-domain transfer method. While these methods have addressed the multi-label issue in medical image recognition, they have not adequately considered the long-tail distribution.

Other existing methods are dedicated to tackling the long-tail distribution problem in medical image recognition, which can be broadly categorized into re-sampling and re-weighting strategies. Re-sampling methods[2, 3, 28, 33] aim to create a balanced dataset by under-sampling the head categories and over-sampling the tail categories, thereby assisting the model in better learning the less common categories. Re-weighting approaches[24, 37, 48] address the long-tail issue by adjusting the loss function weights for different categories. Specifically, these methods assign higher weights to the tail categories and lower weights to the head categories, encouraging the model to focus more on the tail categories. However, the aforementioned methods are not applicable to data with multi-label long-tail distribution. Specifically, re-sampling methods struggle to balance sampling across categories due to the influence of label co-occurrence, while re-weighting methods are highly sensitive to weight parameters in a multi-label context, impacting the model's generalization ability and robustness.

To address the challenges mentioned above, in this paper, we propose a two-stage model based on label decoupling and reconstruction (LDRNet), using class-aware features to reconstruct the distribution of label feature and training an unbiased classifier, to solve the multi-label medical image recognition problem under long-tail distribution. Specifically, in the first stage, we introduce a label decoupling technique that employs prompts and a Self-Attention-based method to decouple class-aware features from the global image features, and training a biased classifier. In the second

stage, we begin by assuming that each category follows a Gaussian distribution, and then reconstruct the Gaussian distribution for each category by the class-aware features obtained in the first stage. Next, we employ a reverse sampling strategy to sample from the Gaussian distributions of each category and generate virtual global image features. Finally, we fine-tune the biased classifier with the generated virtual features to correct its bias. Since our method samples from Gaussian distributions for each category, it generates more diverse features for tail categories, further reducing the classifier's overfitting to tail categories, and thereby enhancing the model's generalization ability and robustness. Additionally, due to the scarcity of publicly available multi-label long-tail distribution medical image datasets, we have collected a long-tail distribution multi-label medical image dataset on Axial Spondyloarthritis to foster development in this field, which we plan to make open source in the future. We evaluated our method on this dataset and two publicly available datasets, and the experimental results demonstrate the superiority of our approach. Our contributions can be summarized as follows:

- We propose a LDRNet for long-tail multi-label medical image recognition, which operates through a two-stage training process. In the first stage, we decouple features for each label from the global image, and in the second stage, we reconstruct the feature distribution for each label, then using reverse-sampling to generate more diverse virtual features to obtain the unbiased classifier.
- We have collected an Axial Spondyloarthritis Dataset to promote the development of the field of multi-label image recognition under long-tail distribution.
- We conducted experiments on our collected dataset and two other publicly available multi-label medical datasets and compared them with current state-of-the-art methods. The final experimental results prove the effectiveness of our method.

## 2 Related Work

### 2.1 Multi-label Medical Images

Multi-label classification tasks have now been extensively studied for predicting multiple category labels [8, 27, 29, 42, 51]. Recently, in the medical image recognition field, to learn the common relations among labels in multi-labels, some methods can be divided into these two major categories: 1) Architecture-Network-based methods, which mainly focus on designing various network variants to learn the commonalities between labels. For example, MLGCN [8], introduces a GCN-based multi-label classification model to capture the relationships between labels; C-Tran [22] utilizes the Transformer to exploit complex dependencies between visual features and labels; CheXNet [46], integrates CNN and Transformer modules, where CNN provides rich input through bottom-up feature extraction for the Transformer, and the Transformer guides the feature extraction in CNN with its top-down attention mechanism, to boost performance. 2) Loss-based methods, which mainly involve designing or adjusting the loss function for each label to optimize the model's handling of multi-label images, ensuring it can effectively handle the complexities of label interdependencies and varying error costs. Among these, MLSL [48] designs a softmax-based loss function to reduce intra-label and inter-label ranking errors

during training, thereby directly optimizing the ranking loss and AUC; ASL [40] introduces a new asymmetric loss, acting differently on positive and negative samples to balance probabilities across various samples. Kobayashi et al.[21] proposes a dual-approach loss, namely the multi-label classification loss for each sample and the label classification loss for each class, to simultaneously improve sample-wise and class-wise discrimination performance, thereby enhancing classification accuracy.

## 2.2 Long-tail Distribution of Medical Images

In the medical field, due to the different incidence rates of diseases, medical image datasets often exhibit a long-tail distribution [10, 11], where common cases account for a large number of samples, and rare cases account for a small number of samples. This long-tail distribution leads to model training bias towards head categories while neglecting tail categories, severely impairing the model's diagnostic performance in real-world scenarios. Therefore, balanced recognition of minority categories is necessary to improve diagnostic performance. To address this issue, most previous methods focus on learning each category in a balance manner, these methods can generally be categorized into three aspects. The re-sampling methods, which include under-sampling for head categories [2] and over-sampling for tail categories [33], focus on constructing a balanced training set. While the re-weighting methods [24, 37] adjust the long-tail distribution by assigning different weights to the samples. Furthermore, recent studies [36, 52] have proposed using a two-stage decoupling approach to address the long-tail issue, achieving notable results. However, these methods are solely applicable to single-label tasks and are not suitable for multi-label tasks, particularly in the context of medical image recognition. Specifically, for re-sampling methods, when a medical image includes both head and tail categories, it is not feasible to achieve balanced sampling through under-sampling or over-sampling strategies, making it difficult to directly be applied. For re-weighting methods, they are highly sensitive to weight parameters in a multi-label context, leading to unstable model training and decreased model robustness.

In real clinical scenarios, where there are both commonality among multiple pathological features and rare pathological features, medical datasets are typically both multi-labeled and long-tail distributed. Therefore, we propose a model for multi-label medical imaging under a long-tail distribution, aiming to simultaneously solve the problems of multi-label and long-tail distribution in real-world medical clinical scenarios. Furthermore, this model also enhances the generalization ability and robustness.

## 3 Method

### 3.1 Problem Setup

Given a multi-label dataset $D = \{I, Y\}$ with $C$ categories and $N$ samples, where $I$ represents the input medical images, and $Y = \{y_1, y_2, ..., y_C\}$ represents the image-level label annotations, with $y_c \in \{0, 1\}$ indicating whether category $c$ is present in the image ('1' for presence and '0' for absence). Data in real medical clinical scenarios often exhibit a long-tail distribution, where head categories comprise the majority of the data, while tail categories have very few samples. The presence of a long-tail distribution poses unique challenges for multi-label image classification,

primarily manifested in learning algorithms tending to optimize performance for categories with a large number of samples while neglecting those with fewer samples. Our objective is to learn a function $f(I) = Y'$, to predict the categories of the input image, and this function $f$ is not affected by the long-tail distribution.

### 3.2 Framework Overview

In this paper, we propose a two-stage method, denoted as LDRNet, to address the multi-label recognition task with long-tail distribution in medical images, the overall framework is shown in Fig. 2. In the first stage, we utilize a Transformer encoder and prompt strategy to decouple class-aware features. In the second stage, we first reconstruct the Gaussian distribution for each category using class-aware features, then apply a reverse sampling strategy to sample and generate virtual multi-label image global features from the Gaussian distribution to fine-tune the classifier from the first stage. Our LDRNet aims to achieve balanced learning of each category by the classifier.

### 3.3 Class-aware Feature Extraction

Our aim is to use class-aware features to reconstruct the Gaussian distribution of each category, so we need to obtain the class-aware features. Current mainstream operations for feature decoupling include CAM-based [7] and Self-Attention-based [26] methods. Given the complexity and specialization of medical image data compared to natural images, we adopt the Self-Attention-based method, which has better performance. Specifically, we first use the backbone to obtain global image features, then use learnable query embeddings $Q$ and a Transformer encoder to decouple the class-aware features from the global features. Formally, given the input multi-label medical image $I$, and using the backbone to obtain global image features $X \in \mathbb{R}^{D \times H \times W}$ :

$$X = f_{backbone}(I), \tag{1}$$

where $D, H, W$ represent the channels, height, and width of the features, respectively. $f_{backbone}(\cdot)$ represents the learnable neural network, for which we primarily use densenet [19] and resnet [17]. Then, we propose a prompt strategy to decouple the class features. Specifically, we first define the prompting template: "The Categories of the image are {LN}", where "LN" represents the name of the labels, and use CLIP[39] to obtain the word embedding $W \in \mathbb{R}^{M \times D'}$. Then, we use a learnable MLP to map the dimensions of the word embedding to $D$. Next, we define a learnable label embedding $L \in \mathbb{R}^{C \times D}$ as the label query, define $P_{word} \in \mathbb{R}^{1 \times D}$ and $P_{label} \in \mathbb{R}^{1 \times D}$ as the learnable indicator embeddings for the prompting word and label. Finally, we fuse the above embeddings and add the learnable position embedding to obtain the query embedding $Q$ :

$$Q = [W + P_{word} : L + P_{label}] + P_{position} \in \mathbb{R}^{(M+C) \times D}, \tag{2}$$

where $[:]$ denotes the concatenation operation, $P_{position} \in \mathbb{R}^{(M+C) \times D}$ is the learnable position embedding. Once we have the query embedding $Q$, we then use the Transformer encoder to implement the decoupling of the category features. Firstly, we reshape the global image features along the spatial dimensions and obtain the flattened feature:

$$X_f = f_{reshape}(X), \tag{3}$$

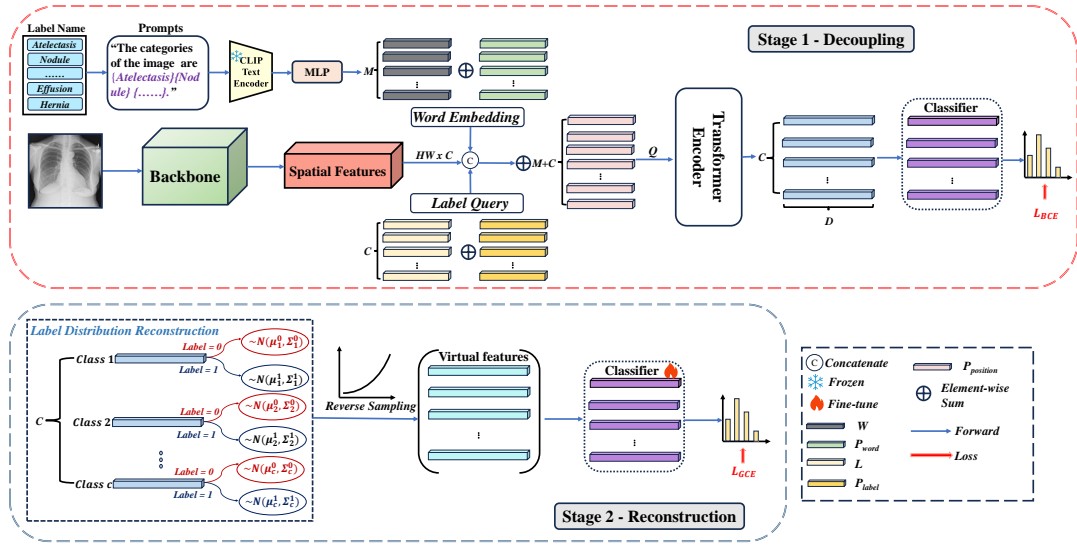

**Figure 2: The overall framework of our proposed method.**

where $X_f \in \mathbb{R}^{HW \times D}$ denotes the flattened feature, and $f_{reshape}(\cdot)$ is the reshape operator. Secondly, we integrate the query embedding $Q$ and flattened feature $X_f$, and use the Transformer encoder to obtain the class-aware features $X_{class}$:

$$X' = f_{encoder}([Q:X_f]), \tag{4}$$

$$X_{class} = f_{select}(X') \in \mathbb{R}^{C \times D}, \tag{5}$$

where $f_{encoder}(\cdot)$ denotes a standard Transformer encoder, which contains a self-attention module and a feed-forward network. $f_{select}(\cdot)$ is the feature selection operation, which selects features from $X'$ at positions corresponding to the label embedding $L$. Then, we can obtain the prediction confidence of class-aware feature $\hat{y}$:

$$\hat{y} = f_{cls}(X_{class}), \tag{6}$$

where $f_{cls}(\cdot)$ represents our classifier, and it tends to exhibit bias. Finally, we employ a Binary Cross-Entropy (BCE) loss to constrain feature learning and achieve the disentangling operation:

$$L_{BCE} = -\frac{1}{N}\frac{1}{C}\sum_{c=1}^{N}\sum_{c=1}^{C}\left[y_{i,c}\log(\sigma(\hat{y}_{i,c})) + (1-y_{i,c})\log(1-\sigma(\hat{y}_{i,c}))\right], \tag{7}$$

where $\sigma(\cdot)$ is the sigmoid function, $y_{i,c}$ is the ground truth label of the $c$-th category for the $i$-th sample, and $\hat{y}_{i,c}$ is the predicted logit for the $c$-th category of the $i$-th sample.

## 3.4 Label Distribution Reconstruction

Unlike long-tail distribution single-label image recognition tasks, multi-label image recognition cannot employ re-sampling methods to balance the sampling of each category due to the influence of label co-occurrence relationships. Inspired by [1, 36], we assume that the feature distribution of each category follows a Gaussian distribution, and propose a Gaussian distribution reconstruction method for multi-label medical imaging under long-tail distribution. This method attempts to reconstruct the Gaussian distribution of each label and uses a reverse sampling strategy to sequentially sample features from the Gaussian distributions to generate a virtual

global feature of the image. This generated feature is then used to fine-tune the classifier trained in the first stage, thereby "correcting" the classifier's bias. Specifically, for the class-aware features $X_{class}^i \in \mathbb{R}^{C \times D}$ of the $i$-th image, we divide along the channel dimension $D$ into feature vectors for each category :

$$X_{class}^i = \{X_{class}^{i,1}, X_{class}^{i,2}, \ldots, X_{class}^{i,C}\}, \tag{8}$$

where $X_{class}^{i,c} \in \mathbb{R}^{1 \times D}$ represents the decoupled feature of the $c$-th category for the $i$-th image.

Then, we construct a corresponding Gaussian distribution for each category. Unlike single-label images, since multi-label images contain multiple categories without location annotation information, decoupling processes may still yield class-aware features at corresponding positions for categories not present in the image. Simply reconstructing features for categories present in the image would disrupt the original feature distribution. Therefore, we reconstruct both label-0 and label-1 Gaussian distributions for each category.

Formally, given the class-aware feature $X_{class}^{i,c}$ for the $c$-th category of the $i$-th image, we calculate the mean and variance for the $c$-th category as follows:

$$\mu_c^j = \frac{1}{N_c^j}\sum_{i=1}^{N_c^j}X_{class}^{i,c}, \tag{9}$$

$$\Sigma_c^j = \frac{1}{N_c^j - 1}\sum_{i=1}^{N_c^j}(X_{class}^{i,c} - \mu_c^j)^T(X_{class}^{i,c} - \mu_c^j), \tag{10}$$

where $j \in \{0, 1\}$ corresponds to label 0 or 1. $N_c^j$ represents the number of samples for category $c$ with label $j$, and $\mu_c^j$ and $\Sigma_c^j$ represent the mean and variance of features for category $c$ with label $j$. Then, based on the calculated mean and variance, we construct the corresponding Gaussian distributions, whose probability density

functions can be written as:

$$f_c^j(X_c^j) = \frac{1}{\Sigma_c^j \sqrt{2\pi}} e^{-\frac{(X_c^j - \mu_c^j)^2}{2(\Sigma_c^j)^2}}, \tag{11}$$

where $j \in \{0, 1\}$ corresponds to label 0 or 1, $X_c^j$ represents the class-aware feature vector with category $c$ label $j$, and $f_c^j(\cdot)$ represents the probability density functions for category $c$ with label $j$. Next, we employ a reverse sampling strategy for each category to achieve feature re-sampling. The sampling probability for each category is as follows:

$$p_c = \frac{\left(\frac{1}{F_c + \epsilon}\right)^\alpha}{\sum_{j=1}^C \left(\frac{1}{F_j + \epsilon}\right)^\alpha}, \tag{12}$$

where $p_c$ denotes the probability of category $c$, $F_c$ represents the frequency of the $c$-th category samples, $\epsilon$ is a constant for numerical stability which is set to 1e-9, and $\alpha$ is a hyperparameter that controls the smoothness, which is set to 0.9. Our ablation study will later show the impact of different values of $\alpha$ on the sampling probability. Then, for each category, we generate independent random numbers $r_c$ to obtain the sampled features, which can be written as:

$$\hat{X}_{class}^c = \begin{cases} X_c^1, X_c^1 \sim \mathcal{N}(\mu_c^1, \Sigma_c^1) & \text{if } r_c \leq p_c \\ X_c^0, X_c^0 \sim \mathcal{N}(\mu_c^0, \Sigma_c^0) & \text{if } r_c > p_c \end{cases} \tag{13}$$

where $\hat{X}_{class}^c \in \mathbb{R}^{1 \times D}$ represents the virtual class-aware feature of the $c$-th category sampled from the Gaussian distribution. Finally, we concatenate all sampled features to obtain the final virtual class-aware feature $\hat{X}_{class}$:

$$\hat{X}_{class} = [\hat{X}_{class}^1 : \hat{X}_{class}^2 : ... : \hat{X}_{class}^C] \in \mathbb{R}^{C \times D}, \tag{14}$$

However, since the Gaussian distribution is computed based on the statistical measures of the feature space from the first stage, to further mitigate potential biases, we employ the Expectation-Maximization algorithm [32, 36] to iteratively fine-tune the classifier and encoder. In the maximization step, we freeze the encoder and backbone, and train the classifier on a balanced feature space. In the expectation step, we freeze the classifier, and supervise the encoder and backbone with an additional balance constraint to avoid being contaminated by the long-tail label space again. Therefore, we use BCE to fine-tune the classifier in the maximization step and design a Generalized Cross Entropy (GCE) loss [50] suitable for multi-label in the expectation step, which can be written as:

$$\hat{y}' = f_{cls}(\hat{X}_{class}) \in \mathbb{R}^C, \tag{15}$$

$$\mathcal{L}_{GCE} = \frac{1}{N}\frac{1}{C}\sum_{i=1}^N \sum_{c=1}^C \left[ -y_{i,c}\left(1 - (\hat{y}_{i,c}')^q\right) + (1 - y_{i,c})(\hat{y}_{i,c}')^q \right]^{1/q}, \tag{16}$$

where $f_{cls}(\cdot)$ represents our classifier trained in the first stage, which we fine-tune in the second stage. The hyperparameter $q$ is in the range $(0, 1]$ and controls the generalization of the loss function. When $q$ is set to 1, the loss function becomes the traditional BCE Loss. In our setup, we set $q$ to 0.8, which our subsequent ablation study shows yields optimal performance. $y_{i,c}$ is the ground truth label for the $c$-th category of the $i$-th sample, and $\hat{y}_{i,c}'$ is the predicted logit for the $c$-th category of the $i$-th sample. Through our reconstructed label distribution, the classifier can classify within a

**Table 1: Comparisons with SOTA on Axial Spondyloarthritis Dataset.**

| Method | AUC | mACC | mAP |
|---|---|---|---|
| DenseNet | 0.7268 | 0.7088 | 0.6059 |
| ResNet | 0.7411 | 0.7087 | 0.6337 |
| MLSL [48] | 0.717 | 0.6854 | 0.6022 |
| Focal Loss [24] | 0.7900 | 0.7610 | 0.7430 |
| Two-way Loss [21] | 0.7439 | 0.7192 | 0.6728 |
| RAL [37] | 0.8010 | 0.68 | 0.7225 |
| DenseNet + LDRNet | 0.8051 | 0.7567 | 0.7510 |
| ResNet + LDRNet | **0.8292** | **0.7838** | **0.7596** |

balanced feature space while learning diverse feature representations, enhancing generalization and robustness, especially for those tail categories with scarce sample sizes.

## 4 Experiments

### 4.1 Datasets

1) **Axial Spondyloarthritis Dataset.** This dataset comprises MRI scans of the sacroiliac joints of 996 patients, with each patient providing between 16 and 24 image slices. Each image slice includes 12 disease image labels: Erosion Right/Left (ER/EL), Sclerosis Right/Left (SR/SL), Joint Space Right/Left (JSR/JSL), Osteophyte Right/Left (OR/RL), Lipomatosis Right/Left (LR/LL), and Bone Marrow Edema Right/Left (BMER/BMEL). These labels were determined by 3 musculoskeletal radiologists (two with 8 years of experience, one with 5 years of experience) and 2 rheumatology specialists (one with 21 years of experience, one with 11 years of experience) based on additional CT images of the patients, assessing structural lesions of the sacroiliac joints on both sides, including erosion, sclerosis, space, and osteophytes. In cases of discrepancy, a thorough discussion was conducted among the five doctors to reach a consensus. Due to a long-tail distribution in the dataset, the head categories include BMER, BMEL, JSR, and JSL, with the remaining categories considered as tail categories.

2) **NIH Chest X-ray 14 Dataset.** The NIH Chest X-ray 14 dataset[44] encompasses 112,120 frontal chest X-ray images from 30,805 patients, each annotated with 14 disease image labels mined through natural language processing from the corresponding radiological reports. The head categories are Pneumothorax, Infiltration, Atelectasis, Emphysema, Edema, Effusion, and Pneumonia, with the remaining categories classified as tail categories.

3) **ODIR-5K Dataset.** The ODIR-5K [23] dataset, comprises a multi-label image dataset with 7,000 images that include patient-level annotations and image-level diagnostic keywords. The dataset encompasses a range of eye diseases, including Normal (N), Diabetes (D), Glaucoma (G), Cataract (C), Age-related Macular Degeneration (AMD), Hypertension (H), Myopia (M), and Others. Among these, the head categories, consisting of N, D, and O, while the remainings are tail categories.

For all datasets, we randomly split the dataset into training, validation, and test sets by a ratio of 7:1:2 at the patient level. Subsequently, we evaluate our methodology on the test set.

**Table 2: Comparisons with SOTA on NIH Chest X-ray 14 Dataset.**

| Method | Number of labels | Wang [44] | DNetLoc [14] | Liu [25] | GWSA+LCD [47] | Xi [35] | PCSANet [5] | A3Net [43] | ImageGCN [30] | MFCNet [4] | ThoraX-PriorNet [18] | DGFN [12] | LDRNet |
|---|---|---|---|---|---|---|---|---|---|---|---|---|---|
| Atelectasis | 7996 | 0.700 | 0.767 | 0.773 | 0.770 | 0.770 | 0.807 | 0.779 | 0.802 | **0.833** | 0.827 | 0.8178 | 0.830 |
| Cardiomegaly | 1950 | 0.810 | 0.883 | 0.889 | 0.877 | 0.870 | 0.910 | 0.895 | 0.894 | 0.915 | 0.902 | **0.9284** | 0.915 |
| Effusion | 9621 | 0.760 | 0.828 | 0.821 | 0.827 | 0.830 | 0.879 | 0.836 | 0.874 | 0.884 | **0.884** | 0.8751 | 0.881 |
| Infiltration | 13914 | 0.660 | 0.709 | 0.710 | 0.701 | 0.710 | 0.698 | 0.710 | 0.702 | 0.717 | 0.723 | **0.7452** | 0.724 |
| Mass | 3988 | 0.690 | 0.821 | 0.829 | 0.821 | 0.830 | 0.824 | 0.834 | 0.843 | 0.853 | **0.867** | 0.8803 | 0.859 |
| Nodule | 4376 | 0.670 | 0.758 | 0.770 | 0.790 | 0.790 | 0.750 | 0.777 | 0.768 | 0.803 | 0.807 | 0.7865 | **0.826** |
| Pneumonia | 978 | 0.660 | 0.731 | 0.713 | 0.732 | **0.820** | 0.750 | 0.737 | 0.715 | 0.770 | 0.764 | 0.7791 | 0.783 |
| Pneumothorax | 3705 | 0.800 | 0.846 | 0.869 | 0.870 | 0.880 | 0.850 | 0.878 | **0.900** | 0.885 | 0.890 | 0.8936 | 0.899 |
| Consolidation | 3263 | 0.700 | 0.745 | 0.749 | 0.746 | 0.740 | 0.802 | 0.759 | 0.796 | 0.810 | 0.812 | 0.8091 | **0.817** |
| Edema | 1690 | 0.810 | 0.835 | 0.847 | 0.847 | 0.840 | 0.888 | 0.855 | 0.883 | 0.893 | **0.908** | 0.8925 | 0.900 |
| Emphysema | 1799 | 0.830 | 0.895 | 0.934 | 0.924 | 0.940 | 0.890 | 0.933 | 0.915 | 0.924 | 0.927 | 0.9397 | **0.947** |
| Fibrosis | 1158 | 0.790 | 0.818 | 0.845 | 0.839 | 0.830 | 0.812 | 0.838 | 0.825 | 0.837 | 0.826 | 0.8175 | **0.849** |
| Pleural Thickening | 2279 | 0.680 | 0.761 | 0.773 | 0.782 | 0.790 | 0.768 | 0.791 | 0.791 | 0.784 | **0.813** | 0.8137 | 0.808 |
| Hernia | 144 | 0.870 | 0.896 | 0.925 | 0.921 | 0.910 | 0.915 | 0.938 | 0.943 | 0.905 | 0.905 | 0.9215 | **0.943** |
| Mean AUC | - | 0.7450 | 0.8070 | 0.8180 | 0.8181 | 0.8190 | 0.8250 | 0.8260 | 0.8320 | 0.8440 | 0.8467 | 0.8501 | **0.8560** |

## 4.2 Evaluation Metrics

In the Axial Spondyloarthritis Dataset, we evaluate our method using three metrics: Area Under the Curve (AUC), mean Accuracy (mACC), and mean Average Precision (mAP), where "mean" denotes the average across all categories. For the NIH Chest X-ray 14 dataset, to fairly compare with current state-of-the-art (SOTA) methods [4, 5, 12, 14, 18, 25, 30, 35, 43, 44, 47], we adopt AUC as the measurement standard. For the ODIR-5K Dataset, in order to compare with current SOTA methods [13, 15, 16, 20, 23, 24, 34], we follow the unified official evaluation metrics proposed by [23], which include Cohen's kappa coefficient, F1 score, and AUC, as well as the Final Score as evaluation metrics.

## 4.3 Implementation Details

During the training phase, we utilized ImageNet-pretrained models of ResNet-50 or DenseNet-121 as the backbone, depending on the specific dataset and objectives. The learning rate was set at $1e-4$, with beta parameters for the Adam optimizer configured at $(0.9, 0.999)$. A weight decay of $1e-5$ was applied, and the batch size was set to 32 to balance computational efficiency and memory usage. Additionally, we adopted a reverse sampling strategy, with parameters $\alpha$ and $\epsilon$ in Eq.12 set to 0.9 and $1e-9$, respectively. For our loss function in Eq. 16, the parameter $q$ was set to 0.8. For input image processing, in the Axial Spondyloarthritis Dataset, images were first resized to dimensions of $256 \times 256$, followed by a cropping step of fixed size $224 \times 224$. In contrast, for the NIH Chest X-ray 14 and ODIR-5K Datasets, to maintain consistency with previous SOTA works in terms of input size, the input images were initially resized to $512 \times 512$ before being cropped to $448 \times 448$.

## 4.4 Comparisons With SOTA Methods

*4.4.1 Comparisons on the Axial Spondyloarthritis Dataset.* We first compared our method with some SOTA methods on the Axial Spondyloarthritis Dataset. As shown in Table 1, this comparison was based on unified data augmentation and data partitioning standards, ensuring the fairness and accuracy of the evaluation. It is evident that our method outperforms previous methods across all evaluation metrics. For instance, our method surpassed the baseline (DenseNet) by 7.8%, 4.8%, and 14.5% in AUC, mACC, and mAP metrics, respectively. Under ResNet, the improvement was 8.8%, 7.5%, and 12.6%. This indicates that our method, through decoupling features and reconstructing label distributions, addresses the bias of classifiers under long-tail distributions, thereby enhancing classifier performance. Compared to other SOTA methods, such as the Two-way Loss [21] with ResNet50 as backbone, our method still excelled by 8.5%, 6.5%, and 8.7% in AUC, mACC, and mAP metrics, respectively. This result further validates the effectiveness of our approach.

*4.4.2 Comparisons on NIH Chest X-ray 14 Dataset.* We also evaluated our proposed model with SOTA methods on the publicly available NIH Chest X-ray 14 dataset. The results are shown in Table 2. Our method achieved a competitive mean AUC score of 0.8560 across 14 thoracic diseases, comparing favorably with previous methods. Moreover, our method showed significant improvement in performance on tail classes. For example, with Nodule pathology, which is challenging to detect and rare in the dataset, other methods struggle to perform well due to the long-tail distribution. However, our proposed method of label decoupling and reconstruction enables the generation of diverse features for the classifier to learn, thereby better detecting such pathologies to aid diagnosis and treatment.

*4.4.3 Comparisons on ODIR-5K.* Similar to the previous datasets, our method outperformed previous approaches on the ODIR-5K dataset, which is shown in the Table 3. Specifically, our Kappa score exceeded other baselines, demonstrating that our model can provide reliable and accurate predictions across various categories, even in cases of uneven data distribution. The improvement in the F1 score indicates our model's success in balancing precision and recall, which is particularly important for medical image classification. A

**Table 3: Comparisons with SOTA on ODIR-5K dataset.**

| Method | Kappa | F1 | AUC | Final Score |
|---|---|---|---|---|
| Li et al. [23] | 0.3072 | 0.8495 | 0.8306 | 0.6624 |
| Jordi et al. [20] | 0.426 | 0.850 | 0.805 | 0.693 |
| Gour and Khanna [13] | 0.433 | 0.853 | 0.849 | 0.712 |
| He et al. [15] | 0.520 | 0.886 | 0.903 | 0.770 |
| BFENet [34] | 0.535 | 0.892 | 0.912 | 0.780 |
| Focal Loss [24] | 0.625 | 0.895 | 0.930 | 0.817 |
| SCFKD [16] | 0.635 | 0.911 | 0.927 | 0.824 |
| LDRNet | **0.663** | **0.917** | **0.934** | **0.838** |

high F1 score means the model does not miss many cases while also maintaining a low false-positive rate. The high AUC score showcases our model's ability to differentiate between different pathological categories, proving its effectiveness in distinguishing between healthy and pathological samples and reducing the risk of false positives and negatives. This is crucial for improving diagnostic accuracy and treatment outcomes for patients. Furthermore, the leading final score further validates our method's superior overall performance.

## 4.5 Ablation Studies

*4.5.1 Effects of Different Stage Processes.* We performed related ablation studies on each stage. The experiments are shown in Table 4, where '+' denotes the inclusion of an element, while '-' indicates its absence. From the Table 4, we can observe that the learning of class-aware features in our first stage shows significant performance improvements over the baseline across three metrics. This indicates that our proposed feature learning strategy effectively captures the global features of complex medical images and decouples them, providing a more robust feature representation for subsequent classification tasks. The enhancement of global features is particularly crucial for medical images, which are subtle and varied. It helps the model to maintain a high recognition rate for common diseases while also improving the ability to recognize rare diseases. Furthermore, the performance significantly improves after using feature decoupling along with label distribution reconstruction. This demonstrates that our method of label distribution reconstruction effectively balances the multi-label medical images under a long-tail distribution, allowing the classifier to learn about each category in a balanced manner. Through this reconstruction, our model can provide a more balanced learning environment when facing imbalanced data distribution, ensuring that each category receives adequate attention. Especially for categories with scarce samples, this approach can significantly enhance the model's ability to learn their features and improve recognition accuracy. Additionally, we observe further improvements in our model's performance across three metrics with the addition of Prompt. This is attributed to the semantic information of labels aiding in the decoupling of features by category. By integrating rich contextual and descriptive information with image data, the Prompt module offers semantic guidance to the model, promoting more accurate recognition of rare and difficult-to-distinguish categories.

*4.5.2 Effects of Different Language Models on Prompt Strategy.* In this section, we explore ablation experiments on different language

**Table 4: Effects of different stage processes.**

| Stage-1 Decoupling | Stage-2 Reconstruction | Prompt | AUC | mACC | mAP |
|---|---|---|---|---|---|
| - | - | - | 0.7411 | 0.7087 | 0.6337 |
| + | - | - | 0.7720 | 0.7530 | 0.7253 |
| + | + | - | 0.8101 | 0.7730 | 0.7522 |
| + | + | + | **0.8292** | **0.7838** | **0.7596** |

models, considering three distinct types of models. For a fair comparison, all models were trained in a two-stage process, with other settings kept consistent. As shown in Table 5, among the three models, the CLIP model demonstrated the best performance, outperforming the other two models by approximately 1.1% in AUC, 0.9% in mACC, and 0.5% in mAP. We believe this may be due to the word embeddings from CLIP possibly containing relevant clinical semantics such as sclerosis and steatosis, enabling the model to utilize more accurate semantic information to facilitate the decoupling of image labels. Moreover, CLIP is trained on image-text pairs, giving it an advantage in integrating textual semantic information with image information.

**Table 5: Effects of different language models.**

| Language Models | AUC | mACC | mAP |
|---|---|---|---|
| Word2vector [31] | 0.8113 | 0.7758 | 0.7540 |
| BlueBERT [38] | 0.8183 | 0.7742 | 0.7570 |
| CLIP [39] | **0.8292** | **0.7838** | **0.7596** |

*4.5.3 Effects of Different Resampling Strategies.* We present the ablation experiments for different sampling strategies in the second stage as depicted in Eq. 12. As shown in the Table 6, we observe that sampling from the multi-label reconstructed Gaussian distribution using our reverse sampling strategy outperforms uniform sampling from the same distribution by approximately 2.9%, 1.9%, and 3.2% across three metrics, respectively. We speculate that since the classifier learned in the first stage already exhibits a significant bias, it is necessary to employ a reverse sampling strategy to eliminate this bias, thereby enabling the classifier to distinguish between categories more equitably. Notably, even when our method employs a uniform sampling strategy in its second stage, the results surpass both the baseline and the performance using only the first stage. We attribute this to our proposed reconstruction of the multi-label Gaussian distribution. Even though uniform sampling on a long-tailed distribution might result in fewer tail samples compared to head samples, our reconstructed multi-label Gaussian distribution generates a rich variety of features for tail categories.

*4.5.4 Effects of Different Hyperparameters.* In our model's hyperparameter ablation experiments, we first focused on the smoothing parameter $\alpha$ within the sampling probability equation in Eq.12. We set the range of $\alpha$ values from 0.1 to 1.0, increasing in steps of 0.1, and conducted experiments across three metrics using a uniform ResNet50 as the backbone on our Axial Spondyloarthritis Dataset, with all other settings kept cxwonsistent. The results, as shown in Fig.3a, indicate that the optimal setting for $\alpha$ is 0.9. This finding suggests that a higher $\alpha$ value helps the model to more

**Table 6: The effect with different Sampling Strategies.**

| Sampling Strategies | AUC | mACC | mAP |
|---|---|---|---|
| ResNet | 0.7411 | 0.7087 | 0.6337 |
| Uniform Sampling | 0.7998 | 0.7650 | 0.7277 |
| Reverse Sampling | **0.8292** | **0.7838** | **0.7596** |

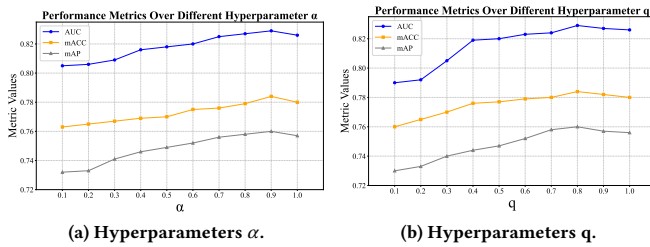

(a) Hyperparameters $\alpha$.  (b) Hyperparameters q.

**Figure 3: Performance evaluation over different hyperparameters $\alpha$ and $q$.**

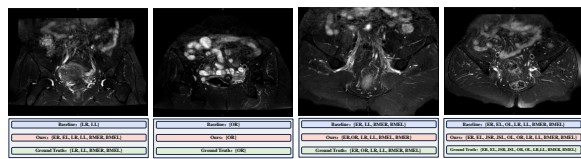

**Figure 4: The first line of each prediction represents the baseline predictions, the second line represents our predictions, and the third line represents the ground-truth.**

balancedly handle categories, especially in situations of class imbalance, thereby enhancing the overall performance of the model.

The parameter $q$ in Eq. 16 controls the generality of the loss function, significantly impacting the model's optimization. By keeping other model parameters constant, we explored a range of $q$ values, set as {0.1, 0.2, ..., 1.0}. As depicted in Fig. 3b, the result indicates that the optimal setting for $q$ is 0.8. This demonstrates that moderately adjusting the generality of the loss function on our dataset can effectively improve the model's generalization capability, thereby optimizing overall performance.

## 4.6 Visualization

In the previous sections, we have conducted some quantitative analyses of our method. In this section, we will carry out a qualitative analysis by performing some visualization experiments on the Axial Spondyloarthritis dataset. As illustrated in Fig. 4, it is observable that our method predicts all pathological features for these medically challenging images, whereas the baseline only predicts pathological features of medical images containing a few labels, such as shown in row 1 column 2. Notably, for the sample in row 1 column 1, our method predicted two additional pathological features, 'ER' and 'EL', beyond the ground truth. We believe this may be due to the complexity of medical images and noise present in the images.

Furthermore, to explore whether our classifier addresses the issue of long-tail distribution, we visualized the L2-Normalization of the parameters in our fine-tuned and baseline classifier. As shown

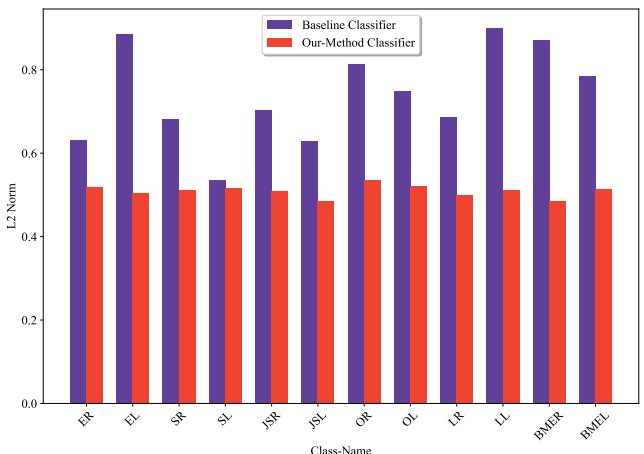

**Figure 5: The results of L2 Normalization of the parameter vectors corresponding to each category in baseline and LDR-Net classifiers.**

in Fig. 5, for the baseline classifier, the influence of the long-tail distribution leads to a bias towards head categories, and the parameters of the classifier are fluctuating. In contrast, the fine-tuned classifier is no evident bias towards any category, and the results are quite stable. This indicates that our method, through feature decoupling and label reconstruction, effectively solves the problem of classifier bias in multi-label medical images under a long-tail distribution, enabling the classifier to treat each category more fairly.

## 5 Conclusion

In this paper, we introduce a novel two-stage training framework, namely decoupling and reconstruction, to effectively address the inherent long-tail distribution problem in multi-label medical image classification tasks. In the first stage, our method initially employs a technique that combines prompt-based approaches with the Transformer model to decouple specific category features from global image features. In the second stage, we reconstruct Gaussian distributions for each category, and then use reverse-sampling to generate more diverse virtual features to correct the biased classifier. Furthermore, acknowledging the scarcity of publicly available multi-label medical image datasets under long-tail distribution, we have collected a multi-label medical image dataset on ankylosing spondylitis and plan to make it open-source to foster research development in the related field. The experimental results on this dataset, along with two other public multi-label datasets, validate the effectiveness and superiority of our approach. In the future, we will try to combine two stages into one stage.

## 6 Acknowledgement

This work was supported in part by the National Natural Science Foundation of China (Grant No. 62202337, U2033210, 62076185), in part by the Zhejiang Provincial Natural Science Foundation (Grant No. LDT23F02024F02), and in part by the Fundamental Research Funds for the Provincial Universities of Zhejiang GK239909299001-019.

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
