# OpenReview forum: "Label Decoupling and Reconstruction: A Two-Stage Training Framework for Long-tailed Multi-label Medical Image Recognition"
_acmmm.org/ACMMM/2024/Conference — MM2024 Poster_

### Official Review · Reviewer_KUeh · 2024-05-09

**Rating:** 4
**Confidence:** 4

**Summary:**

This paper addresses challenges of multi-label and long-tailed distributions in medical datasets using label decoupling and reconstruction. The method involves utilizing class-aware features to reconstruct label distributions and training an unbiased classifier. By decoupling features in the first stage and reconstructing Gaussian distributions in the second stage, the approach aims to improve model generalization and robustness. The study showcases the effectiveness of the proposed method through experiments on various medical image datasets.

**Strengths:**

The approach enhances the model's ability to recognize diseases, captures comprehensive features across various labels, and improves generalization and robustness. The experiments on different medical image datasets show that the model achieves competitive results and outperforms existing methods, particularly in distinguishing between healthy and pathological samples.

**Limitations:**

1. The background research is insufficient. In addition to the unsample method mentioned in the article, there are some other strategies that were ignored by the author of this article, such as
[1]Seo, HyeRyeong, et al. "Enhancing Multi-Label Long-Tailed Classification on Chest X-Rays through ML-GCN Augmentation." Proceedings of the IEEE/CVF International Conference on Computer Vision. 2023.
[2] Zhang, Yilan, et al. "ECL: Class-Enhancement Contrastive Learning for Long-Tailed Skin Lesion Classification." International Conference on Medical Image Computing and Computer-Assisted Intervention. Cham: Springer Nature Switzerland, 2023.
2. At the lines 285-286, transformer-based is not very accurate, because transformer-based is more like a method based on a structure. When introducing the method stream, the main principles should be introduced to avoid misunderstandings.
3. At lines 480-500, why only train the classifier to construct a balanced feature space instead of training the encoder and classifier at the same time? The author should give further explanation
4. The ablation study was not sufficient and was only verified on one data set. In order to fully verify the performance of the method, verification should be done on three data sets.

**Suitability:**

3

---

### Official Review · Reviewer_tLhp · 2024-05-22

**Rating:** 5
**Confidence:** 3

**Summary:**

*Problem statement*: medical datasets often contain data with multiple diseases thus samples show multi-label characteristics, with wildly varying distributions, thus models develop biases towards more frequent labels (obviously).

*Why a naïve solution that doesn’t work*: samples have multiple labels with high co-occurrence, so naïve rebalancing methods cannot increase the low sample size class proportionally to the high sample classes.

*Proposed solution*: transformer and natural language based method to decouple class specific features from global image features. Uses class specific features to learn a Gaussian distribution for those features. Sample features from the learnt distributions to generate “virtual” features, which can then be weighted to balance classes with few samples for unbiased classifier training.
I.E., class specific feature resampling to reduce classifier bias.


Contributions:
1.	New method for rebalancing highly correlated class features
2.	New long tail multiclass medical imaging dataset
3.	Experiments on new dataset and existing datasets, comparison with methods

**Strengths:**

Approach and concept is novel and addresses a real-world problem. The method is mostly described well and appears replicable, with well-defined implementation details. The detailed experiments show the strong performance of the method though some details are missing.

**Limitations:**

*Missing dataset information*: Distribution of class labels not stated, which is the driving force behind the paper. How much overlap between labels is there in each of the datasets? Are the labels independent? Does this matter?

*Method evaluation*: Does the method improve the performance of the tail labels significantly? Or does the accuracy just improve overall?

Demonstrates an improvement in overall accuracy (AUC, mACC, and mAP), but does not demonstrate that it actually impacted the accuracy of imbalanced classes. Does it lead to better accuracy on the “tail classes”, or does it give a blanket improvement in accuracy?

The above points could be addressed by giving the distribution of the class labels for each of the dataset. For table 2, adding a column with the number of samples per class would allow us to see the impact of the method on different label distributions.

Perhaps a naïve question: is it possible to simply train one model per class (i.e., separately) to address the model’s bias? This might inhibit the learning of the model in some way, but then the correlation between classes does not have to be learnt and overcome by the model.

**Suitability:**

2

---

### Official Review · Reviewer_bp8A · 2024-05-25

**Rating:** 4
**Confidence:** 3

**Summary:**

The paper proposes a two-stage framework for long-tailed multi-label medical image recognition. Specifically, it utilizes label decoupling to extract the class-aware features, and then reconstruct the label distribution from class-aware features by reverse sampling strategy and Gaussian distribution reconstruction.

**Strengths:**

1.The paper proposes a two-stage framework for long-tail multi-label medical image recognition, which can capture class-aware features and reconstruct the label distribution to solve the long-tail problem effectively.
2.The paper collect an Axial Spondyloarthritis Dataset to promote the development of the field of multi-label image recognition under long-tail distribution.

**Limitations:**

1.Reference [27] used a transformer decoder to capture class-aware features by cross attention on input features and learnable query. In this paper, the class-aware features are obtained through the concatenation of label query and the features as the input of the transformer encoder, and then the features at the corresponding position are selected to obtain the class-aware features. This approach may seem less intuitive than [27] 's.
2.It is hard to understand the effect of Pword and the Plabel, why not concatenate W and L directly with spatial features directly?
3.Does stage1 and stage2 alternate between each other or does stage2 occur after stage1 has completely ended? If it is alternate, does this mean that Gaussian distribution is built from a batch of data? Does this affect the final result because of the different distribution of the local and the global?

**Suitability:**

2

---

### Meta-Review · Area_Chair_EdL1 · 2024-06-24

**Recommendation:** Accept (Poster)
**Confidence:** 4

**Metareview:**

The paper proposes a two-stage framework for long-tailed multi-label medical image recognition, named Label Decoupling and Reconstruction (LDRNet). The framework aims to address the challenges of multi-label and long-tailed distributions in medical datasets. The first stage involves label decoupling, which extracts class-aware features. In contrast, the second stage involves label reconstruction, using these features to generate diverse virtual features for tail categories, promoting unbiased learning and enhancing model generalization.

Pros:
+ The two-stage label decoupling and reconstruction framework is a novel approach to addressing the long-tail problem in multi-label medical image recognition.
+ The model achieves state-of-the-art performance on multiple datasets, showcasing its effectiveness in handling complex, multi-label, and long-tailed distributions.

Cons:
+ The rebuttal did not significantly solve the problems of the two reviewers.
+ The paper lacks a detailed comparison with existing methods in terms of performance.
+ Further clarification is needed on the detailed process of label decoupling and reconstruction, and the interplay between the two stages.
+ The ablation study is limited to one dataset. A more comprehensive ablation study across all three datasets would validate the method's performance better.

This paper has one weak accept, one borderline accept, and one borderline reject, which is over the bar of acceptance, so I recommend accepting this paper.